# The effect of cognitive function and central nervous system depressant use on mortality—A prospective observational study of previously hospitalised older patients

Tahreem Ghazal Siddiqui [1,2]*, Maria Torheim Bjelkarøy[1,2], Socheat Cheng [1,2], Espen Saxhaug Kristoffersen[1,3,4], Ramune Grambaite[1,5], Christofer Lundqvist [1,2,3]

1 Health Services Research Unit (HØKH), Akershus University Hospital, Lørenskog, Norway, 2 Institute of Clinical medicine, Faculty of Medicine, University of Oslo, Oslo, Norway, 3 Department of Neurology, Akershus University Hospital, Lørenskog, Norway, 4 Department of General Practice, Institute of Health and Society, University of Oslo, Oslo, Norway, 5 Department of Psychology, Norwegian University of Science and Technology, Trondheim, Norway

* tahs@ahus.no

**Data Availability Statement:** Regarding data sharing, we have an ethical restriction from the

## Abstract

### Background

Older patients are often users of prolonged Central Nervous System Depressants (CNSD) (Z-hypnotics, benzodiazepines and opioids), which may be associated with reduced cognition. The long-term effects of CNSD use and reduced cognitive function in older patients are unclear. The aim of this study was to examine whether cognitive function and CNSD use at baseline hospitalisation were associated with all-cause mortality two years after discharge.

### Methods

We conducted a prospective observational study, including baseline data (2017–2018) from previously hospitalised older patients (65–90 years), assessing all-cause mortality two years after discharge. We used logistic regression to assess the primary outcome, all-cause mortality two years after baseline hospitalisation. The primary predictors were cognitive function measured by The Mini Mental State Examination (MMSE) and prolonged CNSD use (continuous use $\geq$ 4 weeks). Adjustment variables: age, gender, education, the Hospital Anxiety and Depression Scale (HADS) and the Cumulative Illness Rating Scale for Geriatrics (CIRS-G), using receiver operating characteristics (ROC) to compare the predictive power of the models. In a sub-analysis we used, the Neurobehavioural Cognitive State Examination (Cognistat) and the Clock Drawing Test.

### Results

Two years after discharge, out of 246 baseline patients, 43 were deceased at follow-up, among these 27 (63%) were CNSD users, and 16 (36%) were non-users at baseline, (p = 0.002). In the multivariable models cognitive function (MMSE score) was a predictor of mortality (OR 0.81 (95% CI 0.69; 0.96), p = 0.014). CNSD use was associated with mortality

regional committees for medical and health research ethics (REC) and Akershus university hospital data protection officer to share data from hospitalised patients. That is due to sensitive information from electronic patient journals with the risk of identifying the individual patient. We have a small sample size in some of the groups in our study, thus making it easier to identify individual patients. For more information, please contact: rek-sorost@medisin.uio.no or personvern@ahus.no.

**Funding:** This work was supported by a grant from the Norwegian Research Council (256431) and the Health Services Research Unit of the Akershus University Hospital. MTB received funding from ELIB (Stiftelsen Et Liv i Bevegelse). The funders had no role in study design, data collection and analysis, decision to publish or preparation of the manuscript.

**Competing interests:** CL has participated on an advisory board and received payment for lectures arranged by Abbvie Pharma AS, Novartis AS and Roche AS, Norway. He has also received research sponsorship from Abbvie pharma, but not for the current project. All other authors declare that they have no conflicts of interest.

(OR 2.71 (95% CI 1.06; 6.95), p = 0.038), with ROC AUC: 0.74–0.77 for these models. Results using Cognistat supported the findings. The Clock Drawing Test was not significant predictor of mortality.

## Conclusion

Two years after discharge from the hospital, older patients with reduced cognitive function and CNSD use during hospital stay had higher mortality. This underlines that inappropriate (prolonged and concurrent) use of CNSDs should be avoided by older patients, particularly in patients with reduced cognitive function.

## Trial registration

NCT03162081, 22 May 2017.

## Introduction

Increase in age is associated with cognitive decline [1]. Survival among older patients is linked to their cognitive status and it is well-established that patients with dementia have higher mortality compared to healthy older adults [2]. Some studies also report increased mortality among patients with mild cognitive impairments [3–5].

Other factors that can shorten life expectancy are comorbid conditions, which may also lead to polypharmacy and is often defined as use of five or more medications [6, 7]. In older adults, the most prescribed medications, for insomnia, anxiety and pain management are central nervous system depressants (CNSD), Z-hypnotics, benzodiazepines (BZD) and opioids [8–11]. Though, these should be used with caution, and in general for short-term use only [12, 13]. CNSDs are often prescribed [14, 15] and many use CNSDs longer than recommended, which may increase the probability of adverse events [16–20].

There are conflicting results on whether older patients exposed to CNSDs over a prolonged periods of time are at risk of increased likelihood of early death or not [21–25]. Some studies have found a combined effect of reduced cognitive function and increased CNSD use leading to reduced survival [22, 26, 27]. However, the relationship between somatic comorbidity, cognitive function and prolonged use of CNSDs is difficult to determine. This is especially related to the link between comorbidity and CNSD use [28], which both affects cognitive function [29, 30]. Overall, further studies are needed to establish whether patients with and without reduced cognitive function, and prolonged use of CNSDs are at risk of increased likelihood of death.

The aim of this study was to examine the effect of cognitive function and CNSD use, on mortality among older patients, over an extended period after discharge from the hospital and being transferred back to primary care. A relevant spectrum of covariates were included, such as age, gender, education, comorbidity and symptoms of depression and anxiety. Moreover, a secondary aim was to assess the effect of other in-depth cognitive tests on mortality.

## Methods

### Design & settings

We conducted a prospective observational study of 246 patients previously admitted to somatic hospital wards of Akershus university hospital, which we have previously addressed in

a cross-sectional study [30, 31]. The hospital has a large inclusion area of both rural and urban area. Mortality was assessed over two years after the discharge. Baseline data was collected during the hospital stay between May 2017 and August 2018. The mortality data was collected with a standardized time of two years after discharge from hospital for the patient.

## Data collection

Mortality data was obtained from hospitals Electronic Patient Records (EPRs) which are continuously updated from the national death registry and the national population registry. Thus, access to EPRs allow estimation of all-cause mortality rates. Beside mortality data, all other data was collected at baseline by the research group. We collected baseline information on: age, gender, education, hospitalisation, CNSD use, comorbidity information and the Mini Mental State Examination (MMSE) score. A sub-sample of patients completed additional cognitive tests at baseline (Cognistat and the Clock drawing test).

The baseline data collection was based on voluntary response sampling. CNSDs information was collected directly from participants, the GPs medication lists and the EPRs at baseline hospitalisation. We collected sociodemographic data, number of days of hospitalisation and clinical data on comorbidity at baseline. All clinical data and measurements were conducted by the first, third and occasionally last author, except routinely collected MMSE and the Clock Drawing Test, which were at times conducted by a trained occupational therapist in the wards. The follow-up data was collected by the second author.

## Participants

The flow chart of participants in the study from baseline to follow-up is shown in Fig 1. All 246 patients from the original cross-sectional study were included. *Inclusion criteria* were: in-patients from the somatic wards of Akershus university hospital Neurology, General Internal Medicine, and/or Geriatric departments between the age of 65 and 90 years old. *Exclusion criteria* were: psychosis, moderate to severe depressive disorder, brain tumour, traumatic brain injury, stroke, and unable to participate due to medical condition. Patients with active delirium during the hospital stay were excluded, as they were unable to complete cognitive examination. We excluded patients with an MMSE score lower than 21 and/or based on medical notes to avoid the inclusion of patients with reduced ability to consent [32]. Patients diagnosed with dementia [33], major neurocognitive disorder [34] and diagnosed moderate to severe depressive episodes were also excluded [35].

## Prolonged use of CNSDs

Prolonged CNSD medication use was defined as using opioid, BZD, Z-hypnotics or a combination of them, regularly ≥4 weeks prior to and during the baseline hospital stay. Non-use was defined as no CNSD use or sporadic CNSD use below the abovementioned threshold.

## Cognitive measures

The MMSE is a screening tool for cognitive impairment [36] which gives an overall score of 30 points, with <25 indicating cognitive impairment [37]. It takes approximately 10–15 minutes to complete. In the current study the range was from 21 to 30, as patients with MMSE below 21 were excluded due reduced ability to consent [38].

The Clock Drawing Test measures executive function, construction, visual-spatial skills and gives a score from 4–5 (normal) and ≤3 (impaired). The Clock Drawing Test is often used together with MMSE [39]. We conducted this test on a sub-sample of the patients.

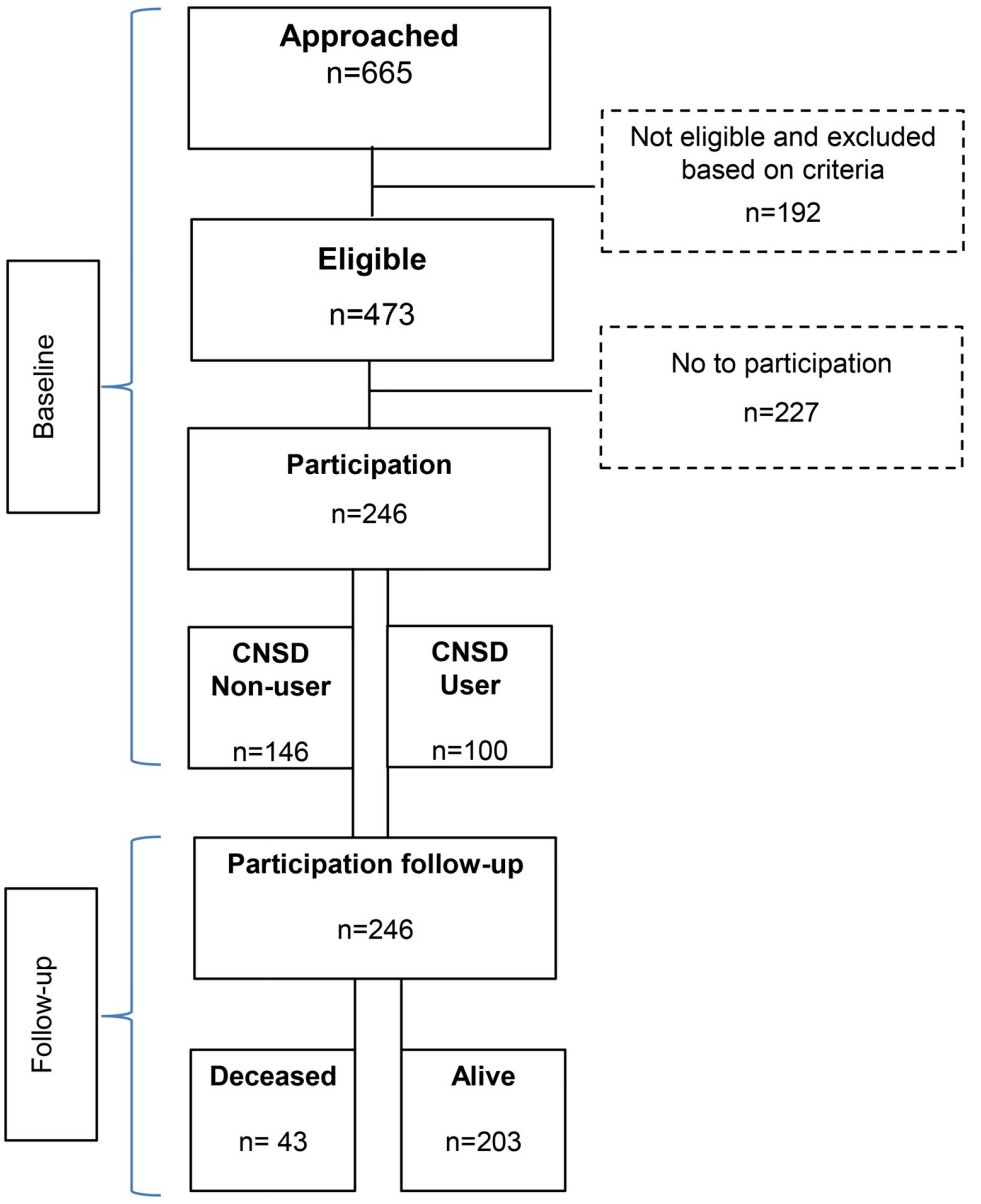

**Fig 1. Flow chart of participation.** Central nervous system depressant.

The Neurobehavioral Cognitive Status Examination (Cognistat) is a multidimensional scale which takes approximately 15 to 20 minutes to perform and addresses general domains (consciousness, orientation and attention) and major domains (language comprehension, memory, construction and reasoning) [40]. We used the total Cognistat score for a subsample of the patients [41].

## Other measures

The Cumulative Illness Rating Score Geriatrics (CIRS-G) score was used to assess burden of disease among individual patients. The scale was used to assess biopsychosocial factors of disease, scoring from no problem to extreme problems (0–4) in major organ systems, such as,

neurological, psychiatric, metabolic and musculoskeletal systems, an increase in score indicated higher burden of disease and comorbidity [42].

The Hospital Anxiety and Depression Scale (HADS) is a 14 items scale, where half of the items represent anxiety and the other half depressive symptoms [43]. The score may be used as a total summary score [43]. We used the total score, with higher score indicating higher levels of symptoms of anxiety and depression.

## Outcome

The primary outcome was all-cause mortality two years after baseline hospital admission.

## Ethics

Participation was by signed informed consent at baseline. The data was stored on a secure hospital research server. The data collection and storage were approved by the Akershus University Hospital data protection officer and the Regional Committees for Medical and Health Research Ethics [2016/2289].

## Statistical analyses

We used IBM SPSS version 23 statistics software [IBM Corp, released 2015 for descriptive analyses, bivariate and multivariate regression. STATA Version 15 [StataCorp, released 2017) was used to conduct the receiver operating characteristic curve (ROC) analysis. The distribution of continuous variables (polypharmacy, Cognistat, the Clock Drawing Test, education in years, age at baseline, HADS, CIRS-G and MMSE) were assessed by graphically inspecting the histograms and were described by means and standard deviation (SD). Categorical variables (CNSD use vs non-use, and gender) were described by frequencies. To assess characteristics and group differences of single variables, significance testing was conducted by using Independent t-test or $\chi^2$ test. P-value limits were set at p≤0.05.

For the primary and secondary analyses, we used multivariate logistic regression models with mortality as primary outcome. The primary variables were: CNSD (use vs. non-use), and MMSE score (range 21–30 as patients scoring <21 were excluded). The included covariates were: age at baseline, gender, education years (as proxy for socioeconomic status) and CIRS-G or HADS, in the models. We used a receiver operating characteristic curve (ROC) analysis to examine the discriminative ability of our multiple logistic regression models. We chose the three models based on our previous cross-sectional study [30]. For the secondary analyses, the included variables were: Cognistat (n = 106) or the Clock Drawing Test (n = 158) and CNSD (use vs non-use), covariates: age, gender, and education.

## Results

### Patients' characteristics

A total of 246 patients were included in the study (Table 1). At two year follow-up, 43 patients were deceased, and among these 27 patients (63%) were CNSD users at baseline, whereas 16 (36%) were non-users. The distribution of medication use among the deceased patients was: Z-hypnotics (n = 16), BZD (n = 0), opioids (n = 4), and n = 7 used a combination of CNSDs (Z-hypnotics + opioids = 4, Z-hypnotics + BZD = 2, all three CNSDs = 1). At baseline, the CNSD use was median 52 weeks (range: 4–988 weeks). The age at baseline, presence of polypharmacy and comorbidity scores were higher among patients deceased after two years. The deceased patients tended to have a longer stay at the hospital at baseline compared to the patients that were alive at follow-up.

**Table 1. Baseline characteristics stratified for alive and deceased patients at two year follow-up.**

| Variables | Deceased at follow-up (N = 43) | Alive at follow-up (N = 203) | Total (N = 246) |
|---|---|---|---|
| Age at baseline[1]* | 80.5 (6.7) | 75.7 (6.3) | 76.6 (6.7) |
| Gender Female, (N/%)[2] | 21 (49%) | 116 (57%) | 137 (56%) |
| Education[2] | 12.6 (3.7) | 13.3 (3.0) | 13.2 (3.1) |
| Days of stay[1]* | 10.5 (6.4) | 7.8 (7.4) | 8.3 (7.3) |
| HADS[2] | 9.9 (6.3) | 8.4 (5.9) | 8.7 (6.0) |
| CIRS-G[1]* | 7.0 (3.0) | 5.6 (2.7) | 5.9 (2.8) |
| Cognistat[1]* | 64.1 (9.0) | 70.2 (6.4) | 69.4 (7.1) |
| Polypharmacy[1]* | 9.4 (4.0) | 7.3 (4.0) | 7.7 (4.1) |
| Clock Drawing Test[1]* | 3.4 (1.6) | 4.1 (1.3) | 3.9 (1.4) |
| MMSE[1]* | 24.1 (2.5) | 25.7 (2.7) | 25.7 (2.7) |
| CNSD use, (N/%) [2]* | 27 (63%) | 73 (36%) | 100 (41%) |
| CNSD non-use, (N/%) | 16 (37%) | 130 (64%) | 146 (59%) |

*p<0.05

1) Independent t-test

2) χ2 test. All data are presented as mean and standard deviation, except for frequencies/% for gender and CNSD. HADS = Hospital Anxiety Depression Scale, CIRS-G = Cumulative Illness Rating Scale Geriatrics, Cognistat = The Neurobehavioral Cognitive Status Examination, MMSE = The Mini Mental State Examination. CNSD = Central Nervous System Depressants. Hospital Anxiety and Depression Scale (HADS) = 17, smoking = 40, education = 10, The Neurobehavioral Cognitive Status Examination (Cognistat) = 140; MMSE Mini mental state examination (MMSE) = 31; Clock drawing test (Clock) = 88).

## Primary outcome

In **Table 2,** both unadjusted and adjusted results are reported. In the unadjusted model, MMSE (OR 0.80 (95% CI 0.70 to 0.92), p = 0.002), CNSDs (OR 3.01 (95% CI 1.52 to 5.94), p = 0.002), age at baseline (OR 1.12 (95% CI 1.06 to 1.19), p<0.001) and CIRS-G (OR 1.17 (95% CI 1.05 to 1.31), p = 0.006) were associated with mortality.

In the adjusted multivariate models, cognitive function measured by MMSE at baseline was associated with mortality (OR 0.81 (95% CI 0.69 to 0.96), p = 0.014). Thus, patients with higher cognitive functions were more likely to survive, in all the models. CNSD use status was

**Table 2. Bivariate and multivariate logistic regression models for mortality, with ROC analysis (AUC).**

| Variables | Bivariate<br>Odds ratio (95% CI), p value | Multivariate model 1<br>Odds ratio (95% CI), p value | Multivariate model 2<br>Odds ratio (95% CI), p value | Multivariate model 3<br>Odds ratio (95% CI), p value |
|---|---|---|---|---|
| MMSE | 0.80 (0.70 to 0.92), p = 0.002* | 0.81 (0.69 to 0.96), p = 0.012* | 0.81 (0.68 to 0.96), p = 0.013* | 0.81 (0.69 to 0.96), p = 0.014* |
| CNSD Non-use (ref) | | - | | |
| CNSD Use | 3.01 (1.52 to 5.94), p = 0.002* | - | 3.04 (1.29 to 7.12), p = 0.011* | 2.71 (1.06 to 6.95), p = 0.038* |
| Gender: Females (ref) | | | | |
| Gender: Males | 1.40 (0.72 to 2.70), p = 0.321 | 1.60 (0.71 to 3.60), p = 0.255 | 1.87 (0.81 to 4.31), p = 0.141 | 1.87 (0.81 to 4.31), p = 0.575 |
| Age at baseline | 1.12 (1.06 to 1.19), p<0.001* | 1.10 (1.03 to 1.17), p = 0.004* | 1.09 (1.02 to 1.16), p = 0.014* | 1.08 (1.01 to 1.16), p = 0.019* |
| Education years | 0.92 (0.82 to 1.03), p = 0.145 | 0.93 (0.87 to 1.13), p = 0.925 | 1.01 (0.88 to 1.15), p = 0.938 | 1.01 (0.88 to 1.16), p = 0.885 |
| CIRS-G | 1.17 (1.05 to 1.31), p = 0.006* | - | - | 1.05 (0.89 to 1.23), p = 0.575 |
| HADS | 1.04 (0.98 to 1.10), p = 0.162 | 1.05 (0.99 to 1.11), p = 0.110 | 1.03 (0.97 to 1.15), p = 0.346 | 1.03 (0.97 to 1.09), p = 0.348 |
| ROC analysis AUC (95% CI) | - | 0.74 (0.65 to 0.82) | 0.77 (0.68 to 0.85) | 0.77 (0.68 to 0.85) |

*p<0.05. MMSE: The Mini Mental State Examination, CIRS-G: Cumulative Illness Rating Scale–Geriatrics. ROC: Receiver Operating Characteristic, AUC: Area Under Curve. CI; confidence interval, Ref: reference category.

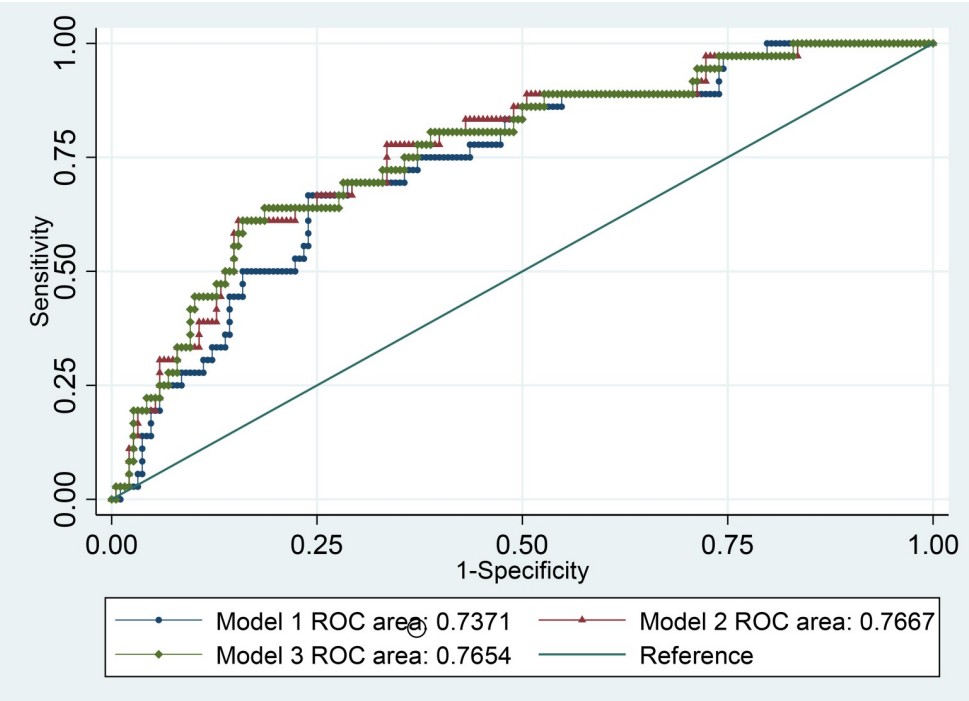

**Fig 2. ROC analysis for multivariate models predicting mortality.**

independently associated with mortality. Patients using CNSD at baseline had increased odds of mortality at follow-up (OR 2.71 (95% CI 1.06 to 6.95), p = 0.038) in all the models.

Age (OR 1.08 (95% CI 1.01 to 1.16), p = 0.019) was also consistently a significant predictor of mortality. Education, HADS, CIRS-G (comorbidity) and gender, were not associated with mortality in the multivariate models.

We used ROC analysis (Fig 2), to compare the three multivariable models, which indicated an area under curve (AUC) 0.74 for model 1, 0.77 for model 2 and 0.77 for model 3. The three models were not significant different in predicting mortality ($\chi^2$ test, p = 0 .588).

## Explorative secondary analyses

In a subsample (n = 106), Cognistat score was the only independent predictor associated with mortality, (OR 0.90 (95% CI 0.82 to 0.98), p = 0.016). Patients with higher Cognistat score were less likely to be deceased at follow-up. In this case we adjusted for the covariates age, gender and education, due to low power, we had to choose fewer covariates). Age was also associated with mortality (OR 1.14 (95% CI 1.02 to 1.27), p = 0.019).

In another subsample (n = 158) with patients who conducted the Clock Drawing Test, the results were not significantly associated with mortality, after adjusting for the same covariates. In this case only age was a significant covariate (OR 1.08 (95% CI 1.02 to 1.18), p = 0.014).

## Discussion

We assessed the effect of reduced cognitive function and prolonged CNSD use on mortality among previously hospitalised older patients. We found that lower cognitive function at baseline hospital admission was an independent predictor of mortality two years after discharge. Moreover, patients using CNSDs at baseline hospitalisation also had a higher mortality two

years after discharge. Age was also associated with higher mortality. Our secondary results support the primary outcome, in indicating that mortality is linked to lower cognitive function on another cognitive measure (Cognistat), although this should be interpreted with caution due to small sample size.

Our findings are in agreement with previous research suggesting increased mortality in patients who have reduced cognitive function [3–5, 21], and among older patients using CNSDs [22, 26, 27]. Nevertheless, one study found no association between mortality and BZD [23]. Another study suggest that patients using Z-hypnotics had a lower risk of mortality than patients using other BZDs [22], which differs from our results. Most of our patients used Z-hypnotics, either alone or in combination with other CNSDs, and very few used BZD alone. This might explain the difference in results. In addition, previous studies also indicate that reduced cognitive function is associated with prolonged CNSD use [30, 44]. Thus, the interaction between CNSDs and cognitive function may partly explain increased mortality after two years in our sample. Previous studies have examined whether the relationship between CNSD use and mortality is causal. One study suggests the possibility of a confounding relationship between CNSD use and mortality, rather than a causal effect of CNSDs on mortality [24]. The results from this study showed an increase in prescription of CNSD due to symptomatic treatment (e.g pain, sleep issues or anxiety) prior to death, and not a causal pathway between CNSD use and death [24]. However, the previous study was a register-based study with data on withdrawals of CNSD prescription. Moreover, this study only adjusted for age and gender. In our study, we included patients from the hospital and examined which CNSDs the patients were using at baseline. Besides, age and gender we also adjusted for comorbidities. Our results indicate that comorbidity was not associated with mortality in the adjusted model. On the other hand, a review that examined Z-hypnotic use and mortality suggest a causal relationship between CNSDs and mortality, via drug effects on physiological mechanisms which may suppress respiration in CNSD users [25]. Unfortunately, our study did not have access to the cause of death for the included patients.

In the adjustment variables, increased age was linked to higher mortality, similar to another study [21], while gender was not a predictor. Length of education was not associated with mortality, but may have affected the performance on cognitive tests [45]. Anxiety and depressive symptoms at baseline were not associated with mortality after two years, others have found contrary results [46]. Anxiety and depression symptoms can also be linked to reduced cognitive function [47]. Nevertheless, we cannot exclude that a later development of anxiety or depression might still play a role in mortality, as we only have baseline data. Comorbidity was not an independent predictor of mortality, which is in contrast to other research [5, 7]. This discrepancy might be related to prolonged CNSD use being associated with comorbidity, [28], but with uncertain causal direction of the relationship. Furthermore, both CNSDs and comorbidity influences cognitive function in older patients [30, 48]. It is noteworthy that including comorbidity in the multivariable regression models, did not improve predictive power of the models, meaning that all three models were similar in predicting mortality.

## Strengths and limitations

We did not reassess the patients after hospitalisation on functional, cognitive status or medication information. Thus, some of the patients could have stopped using CNSDs after the hospitalisation. However, the included patients in the study had previously used CNSD over a median duration of 52 weeks (range: 4–988 weeks) [30], which may suggest it to be unlikely that many would change their medication significantly. We could have included other medication group such as antidepressants, however this was not the focus of this study as we wanted

to assess CNSDs with additive potential that are among the most used in older patients. The sample size limited the number of covariates we could include. Our data is also from a hospital sample, thus the generalisability is limited to hospitalised patients with similar profile as ours. Even so, the patients included were discharged and returned home after hospital stay. They were in general cared for in primary health care both before and after baseline admission. Lastly, we did not have access to the cause of death, this data would be helpful in further understanding the relationship between cognitive function, CNSD use and mortality and should be included in future studies.

On the other hand, the strength of this study is that we had access to well-defined individual data at baseline to assess mortality prospectively over two years and we had no dropouts. Additionally, not many have assessed the concurrent use of BZD, Z-hypnotics and opioids among older patients together with cognitive function, and their combined effect on mortality. In the future, it would be interesting to examine whether a change in CNSD use that also may affect cognitive function, could change mortality among the patients.

## Implications for clinical practise

The current study is close to clinical practise. During discharge from hospital relevant physician should clearly and carefully communicate the patients' treatment plan. Especially, medication information, including adverse effects of CNSDs should be discussed [49]. The concurrent and prolonged use of CNSDs can be evaluated during the stay at hospitalisation, as the hospital has resources to treat older patients with complex conditions. Moreover, regularly conducting collaborative medication reviews with the relevant physicians and general practitioner (GP) is important, as supported by a previous study [50]. The role of GP is important in frequent and careful consideration of pros and cons of CNSDs as GPs often prescribe CNSDs to their older patients [14, 15]. The GPs also have more frequent contact with their patients and can follow their cognitive function and CNSD use.

## Conclusion

Two years after discharge from the hospital, older patients with reduced cognitive function and CNSD had higher mortality. Great care is necessary in the prescription of CNSD to older patients with reduced cognitive function. Our study underlines that inappropriate (prolonged and concurrent) use of CNSDs should be avoided by older patients, particularly in patients with reduced cognitive function. GPs in collaboration with other health care professionals have a central role in providing this support as they might be in regular contact with their patients.

## Acknowledgments

We gratefully acknowledge statistical support of research professor Jūratė Šaltytė Benth. In addition, we appreciate the support during data collection from department secretaries, occupational therapists, physiotherapists, care assistants, nurses and doctors in Geriatric, General Neurology, Internal Medicine and Stroke departments at Akershus University Hospital. We also recognise the extraordinary commitment of patients that participated in this study.

## Author Contributions

**Conceptualization:** Tahreem Ghazal Siddiqui, Maria Torheim Bjelkarøy, Socheat Cheng, Espen Saxhaug Kristoffersen, Ramune Grambaite, Christofer Lundqvist.

**Data curation:** Tahreem Ghazal Siddiqui, Maria Torheim Bjelkarøy, Socheat Cheng, Christofer Lundqvist.

**Formal analysis:** Tahreem Ghazal Siddiqui, Maria Torheim Bjelkarøy, Christofer Lundqvist.

**Funding acquisition:** Espen Saxhaug Kristoffersen, Christofer Lundqvist.

**Investigation:** Tahreem Ghazal Siddiqui, Espen Saxhaug Kristoffersen, Ramune Grambaite, Christofer Lundqvist.

**Methodology:** Tahreem Ghazal Siddiqui, Socheat Cheng, Espen Saxhaug Kristoffersen, Ramune Grambaite, Christofer Lundqvist.

**Project administration:** Tahreem Ghazal Siddiqui, Maria Torheim Bjelkarøy, Socheat Cheng, Christofer Lundqvist.

**Resources:** Socheat Cheng, Ramune Grambaite.

**Supervision:** Espen Saxhaug Kristoffersen, Ramune Grambaite, Christofer Lundqvist.

**Validation:** Socheat Cheng.

**Visualization:** Christofer Lundqvist.

**Writing – original draft:** Tahreem Ghazal Siddiqui.

**Writing – review & editing:** Tahreem Ghazal Siddiqui, Maria Torheim Bjelkarøy, Socheat Cheng, Espen Saxhaug Kristoffersen, Ramune Grambaite, Christofer Lundqvist.

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
