## [Decision Letter · Decision Letter 0]

6 Dec 2021

PONE-D-21-28961The effect of cognitive function and central nervous system depressant use on mortality– a prospective observational study of previously hospitalised older patientsPLOS ONE

Dear Dr. Siddiqui,

Thank you for submitting your manuscript to PLOS ONE. After careful consideration, we feel that it has merit but does not fully meet PLOS ONE’s publication criteria as it currently stands. Therefore, we invite you to submit a revised version of the manuscript that addresses the points raised during the review process.

ACADEMIC EDITOR:

Thank you for your patience while we received reviews on your manuscript. You will see that both reviewers were fairly positive about your manuscript but both also raised issues related to confounding that should be further addressed. The first reviewer raises concerns about multimorbidity and its role in impacting on the association under study but is looking for greater expansion in the Discussion while the second reviewer raises questions about polypharmacy and its treatment in the analysis. Given that these issues (polypharmacy and multimorbidity) are not completely distinct, a more nuanced approach with your manuscript would be helpful.

We look forward to receiving your revised manuscript.

Kind regards,

Andrea Gruneir

Academic Editor

PLOS ONE

Journal Requirements:

Reviewers' comments:

Reviewer's Responses to Questions

**Comments to the Author**

1. Is the manuscript technically sound, and do the data support the conclusions?

Reviewer #1: Yes

Reviewer #2: Yes

2. Has the statistical analysis been performed appropriately and rigorously? 

Reviewer #1: I Don't Know

Reviewer #2: Yes

3. Have the authors made all data underlying the findings in their manuscript fully available?

Reviewer #1: Yes

Reviewer #2: Yes

4. Is the manuscript presented in an intelligible fashion and written in standard English?

Reviewer #1: Yes

Reviewer #2: Yes

5. Review Comments to the Author

Reviewer #1: This is a straightforward article describing prospectively collected information about cognitive functioning and medication use and matching that information to mortality data. Overall, the article is in good shape, but the authors should address a few points:

- page 15 lines 246-249: the manuscript appears to say that the possibility of confounding between CNSD use and mortality is undermined by a previous study's failure to control for comorbidities, but I do not understand this logic. The possibility that CNSD use could overlap with comorbidities that may or may not be accounted for in traditional comorbidity assessments remains an important one to consider--and I would argue that this is among the most important puzzles that the authors are trying to solve. The discussion about this problem should be stronger.

- page 16 line 270: it may be worth reporting (probably in the results section) what the actual range was in terms of duration of CNSD use.

- it should also be acknowledged that we cannot assume that all CNSD impact mortality in the same way, even if there is a reasonable hypothesis that it is through a shared mechanism of respiratory suppression. There is a risk of overgeneralizing the findings, and this is a limitation. It would be worth reporting how many subjects in the CNSD sample were taking which medications exactly, and the 7 combinations among deceased subjects should also be reported.

Reviewer #2: This is a study that well proves the research hypothesis using hospital data. However, there are minor enhancements as follows.

- Please, spell out the abbreviated words below each table

- IRB approval number need to be shown in the ethics part.

- It is necessary to explain in detail what kind of work (ex: data mining) or analysis (ex: multivariate logistic regression model) was performed with which statistical software.

- A clear definition of “polypharmacy” is needed

- There was a difference of frequency in polypharmacy between the two groups (death or alive), but why was it not included as a covariate in multivariate logistic regression?

6. PLOS authors have the option to publish the peer review history of their article (what does this mean?). If published, this will include your full peer review and any attached files.

Reviewer #1: No

Reviewer #2: **Yes: **Sun-Kyeong Park

---

## [Author Response · Author response to Decision Letter 0]

4 Jan 2022

Thank you for taking the time to review our manuscript. We have made changes as suggested in order to improve the manuscript. Please see the response to each question below. 

5. Review Comments to the Author

Reviewer #1: This is a straightforward article describing prospectively collected information about cognitive functioning and medication use and matching that information to mortality data. Overall, the article is in good shape, but the authors should address a few points:

- page 15 lines 246-249: the manuscript appears to say that the possibility of confounding between CNSD use and mortality is undermined by a previous study's failure to control for comorbidities, but I do not understand this logic. The possibility that CNSD use could overlap with comorbidities that may or may not be accounted for in traditional comorbidity assessments remains an important one to consider--and I would argue that this is among the most important puzzles that the authors are trying to solve. The discussion about this problem should be stronger.

We have expended on the discussion on page 15, lines 247-262. 

- page 16 line 270: it may be worth reporting (probably in the results section) what the actual range was in terms of duration of CNSD use.

Thank you for this comment. We have now added this to the manuscript in the result section (page 10 line 192-193) and discussion (page 16 lines, 270). 

- it should also be acknowledged that we cannot assume that all CNSD impact mortality in the same way, even if there is a reasonable hypothesis that it is through a shared mechanism of respiratory suppression. There is a risk of overgeneralizing the findings, and this is a limitation. It would be worth reporting how many subjects in the CNSD sample were taking which medications exactly, and the 7 combinations among deceased subjects should also be reported.

Thank you for the feedback. We have added this to the discussion page 15, lines 247-262. We also added to the limitation section, page 16, line 278-280. We have now reported the combination use of the 7 patients on page 10, line 195-196. 

Reviewer #2: This is a study that well proves the research hypothesis using hospital data. However, there are minor enhancements as follows.

- Please, spell out the abbreviated words below each table

Thank you for this comment, we have added this to page 10, below table 1

- IRB approval number need to be shown in the ethics section

We have now added this to the ethics section, page 8 line 150.

- It is necessary to explain in detail what kind of work (ex: data mining) or analysis (ex: multivariate logistic regression model) was performed with which statistical software.

We have now added this information to page 8, line 152-154

- A clear definition of “polypharmacy” is needed. 

We have now added a definition of polypharmacy in the introduction. Line 48-49, page 4. 

- There was a difference of frequency in polypharmacy between the two groups (death or alive), but why was it not included as a covariate in multivariate logistic regression?

We could not include too many variables in the analysis due to small sample size. Therefore, focused on the comorbidity as main adjustment variable.

---

## [Editor Report · Decision Letter 1]

11 Jan 2022

The effect of cognitive function and central nervous system depressant use on mortality– a prospective observational study of previously hospitalised older patients

PONE-D-21-28961R1

Dear Dr. Siddiqui,

We’re pleased to inform you that your manuscript has been judged scientifically suitable for publication and will be formally accepted for publication once it meets all outstanding technical requirements.

Kind regards,

Andrea Gruneir

Academic Editor

PLOS ONE
---

## [Editor Report · Acceptance letter]

17 Jan 2022

PONE-D-21-28961R1 

The effect of cognitive function and central nervous system depressant use on mortality– a prospective observational study of previously hospitalised older patients 

Dear Dr. Siddiqui:

I'm pleased to inform you that your manuscript has been deemed suitable for publication in PLOS ONE. Congratulations! Your manuscript is now with our production department. 

Kind regards, 

on behalf of

Dr. Andrea Gruneir 

Academic Editor

PLOS ONE